# Complex phase transitions and phase engineering in the aqueous solution of an isopolyoxometalate cluster

Zhi-Da Wang[1,2,6], Song Liang[2,6], Yuqing Yang[3], Zhen-Ning Liu [2], Xiao-Zheng Duan [4] ✉, Xinpei Li[5], Tianbo Liu [3] ✉ & Hong-Ying Zang [1] ✉

Inorganic salts usually demonstrate simple phasal behaviors in dilute aqueous solution mainly involving soluble (homogeneous) and insoluble (macrophase separation) scenarios. Herein, we report the discovery of complex phase behavior involving multiple phase transitions of clear solution – macrophase separation – gelation – solution – macrophase separation in the dilute aqueous solutions of a structurally well-defined molecular cluster $[Mo_7O_{24}]^{6-}$ macroanions with the continuous addition of $Fe^{3+}$. No chemical reaction was involved. The transitions are closely related to the strong electrostatic interaction between $[Mo_7O_{24}]^{6-}$ and their $Fe^{3+}$ counterions, the counterion-mediated attraction and the consequent charge inversion, leading to the formation of linear/branched supramolecular structures, as confirmed by experimental results and molecular dynamics simulations. The rich phase behavior demonstrated by the inorganic cluster $[Mo_7O_{24}]^{6-}$ expands our understanding of nanoscale ions in solution.

Inorganic salts in their dilute aqueous solutions usually demonstrate simple phase transitions—between a homogeneous solution and a macrophase separation (*aka.* soluble and insoluble). The addition of extra electrolytes often decreases the solute's solubility and favors phase separation (precipitation). This common understanding might become invalid when dealing with soluble ions with larger sizes, *e.g.*, on nanometer scale (macroions). Due to their large sizes and multiple charges, macroions are known to moderately attract counterions around the macroions (counterion association) at moderate level. This delicate electrostatic interaction leads to many more fascinating solution behaviors not seen in simple ionic solutions or colloidal suspensions.

Polyoxometalates (POMs) are a type of metal-oxo clusters with inherent charges and large surfaces; many of them stay as macroions with fully hydrophilic surfaces, uniform shapes, and intact molecular structures in dilute solution, which make them ideal models for exploring of the solution properties of macroions[1–4]. The POM clusters, like many macroions, can loosely attract their counterions, selectively favoring those with higher valance or those monovalent ions with smaller hydrated sizes accurately. This counterion association is known to generate counterion-mediated attraction among the POMs and often form 2-D nanolayers. Eventually, the nanosheets will bend and close the edge, forming hollow, spherical, single-layered blackberry structures, when the nanosheets are large enough and their edge energy surpass the bending energy[5–7]. In addition, the macroion-based hydrogels from retained 2-D nanosheets are observed when the nanosheets are further stabilized either by sigma-π interaction or Stronger counterion-mediated attraction[8,9] makes bending more difficult, leading to blackberry structure with larger sizes, or no bending

[1]Key Laboratory of Polyoxometalate and Reticular Science of the Ministry of Education, Faculty of Chemistry, Northeast Normal University, Changchun 130024, China. [2]Key Laboratory of Bionic Engineering (Ministry of Education), College of Biological and Agricultural Engineering, Jilin University, Changchun 130022, China. [3]School of Polymer Science and Polymer Engineering, The University of Akron, Akron, OH 44325, USA. [4]State Key Laboratory of Polymer Physics and Chemistry, Changchun Institute of Applied Chemistry, Chinese Academy of Sciences, Changchun 130022, China. [5]South China Advanced Institute for Soft Matter Science and Technology, Guangdong Provincial Key Laboratory of Functional and Intelligent Hybrid Materials and Devices, South China University of Technology, Guangzhou 510640, China. [6]The authors contributed equally: Zhi-Da Wang, Song Liang. ✉e-mail: xzduan@ciac.ac.cn; tliu@uakron.edu; zanghy100@nenu.edu.cn

at all, leaving the standalone 2-D sheets in solution, which eventually form hydrogels due to their large extended volumes. However, this general trend would not be applicable with poorly charged clusters. With carrying only a few charges, they cannot form 2-D sheets via counterion-mediated attraction, especially with multivalent counter-cations. An important question is whether and how such clusters attract with each other, and if yes, what are the consequent self-assembly and phase behaviors in their aqueous solutions.

Herein, we report the discovery of a homogeneous inorganic hydrogel system with $[Mo_7O_{24}]^{6-}$ (**Mo7**) clusters as the modular units together with multivalent cations, such as $Fe^{3+}$ and $Y^{3+}$. More interestingly, the gel phase belongs to a sequence of complex phase transitions from clear solution to precipitation, to homogenous hydrogel, then to solution phase again, until an eventual precipitation, with the addition of multivalent electrolytes. The distinct phenomena can be attributed to the continuous addition of $Fe^{3+}$ causing charge inversion of **Mo7** macroions, and then followed by a 1-D fiber-like supramolecular structure formation, as confirmed by various techniques and molecular dynamics simulations.

## Results

In a clear, dilute 2.5 mL aqueous solution of $(NH_4)_6[Mo_7O_{24}]$ (0.02 mol/L), precipitation is observed when adding 2.5 mL of $Fe(NO_3)_3$ (0.1 mol/L), indicating that the solubility of **Mo7** decreases with the presence of $Fe(NO_3)_3$ (pH-0.9). With the titration of more volume of $Fe(NO_3)_3$, solid precipitates disappeared and the solution turned to a hydrogel phase. The hydrogel phase can form within the concentration ratio of Mo7:$Fe^{3+}$ ranging from 1:5 to 1:10 (pH-0.3). However, when the concentration ratio of Mo7:$Fe^{3+}$ was 1:5, the gel in olive green color was not transparent. We assumed that part of $Fe^{3+}$ was reduced to $Fe^{2+}$ due to $NH_4^+$. To obtain the homogeneous transparent gel, we further explored more parameters and obtained the transparent yellow gel with Mo7:$Fe^{3+}$ being 1:10. The XPS data for Fe $2p$ orbital is divided into three integrated peaks, the peaks at 712.2 and 726.1 eV belong to Fe $2p_{3/2}$ and Fe $2p_{1/2}$ of $Fe^{3+}$, respectively (Supplementary Fig. 1a). The following characterization was for this transparent gel. The yellow hydrogel was further confirmed by the time-sweep rheology analysis (Supplementary Fig. 1b), as the storage modulus (G′) turned out to be constantly higher than the loss

modulus (G″). Interestingly, when even more $Fe^{3+}$ ions were added, the hydrogel was dissolved again and turned into a clear solution. This upper solution phase covers a broad range of $Fe^{3+}$ concentration in the phase diagram (Fig. 1) until after -3.5 mol/L $Fe^{3+}$ ions were added, phase separation was observed again.

The whole phase transition processes of solution – precipitation – hydrogel – solution – precipitation is very unusual for the aqueous solution of a common soluble inorganic ion. Two issues need much attention in the phase diagram – the gelation and the two separated solution phases at different $Fe^{3+}$ concentrations. A simple estimate on a dilute (0.02–0.16 mol/L) **Mo7** aqueous solution indicates that they would not form gel if they exist as individual clusters due to their long average inter-cluster distance. Gelation requires the formation of certain open, anisotropic supramolecular structures. The question is what driving forces are responsible for triggering the self-assembly and consequently the hydrogel formation. To address this, the electronic and coordination structures of the metal components in Mo7–Fe complexes were explored by using X-ray absorption spectroscopy (XAS). The near-edge absorption energies between dried Mo7–Fe complexes and $Fe_2(MoO_4)_3$ standards (Fig. 2a, c) showed high similarity, proving $Fe^{3+}$ and $Mo^{6+}$ in the gel. Remarkably, the Mo–O–Fe bond was not found in the Mo7–Fe hydrogel[10], illustrating that the hydrogel formation isn't based on covalent bonding between **Mo7** and $Fe^{3+}$ (Fig. 2b and 2d). Meanwhile, the hydrogel is also observed when titrating $Y(NO_3)_3$ into **Mo7** aqueous solution, while $Co^{2+}$ and $Ni^{2+}$ cannot trigger gelation and the aqueous solution remains in its liquor state. As indicated by Coulomb's law, the electrostatic force is proportional to the charge of two ions. We assumed that due to the less charge (compared to $Fe^{3+}$), the divalent $Co^{2+}$ and $Ni^{2+}$ ions cannot form gel with **Mo7** in an aqueous solution.

To understand how the clusters assembled into supramolecular structures in the solution, static light scattering (SLS) and dynamic light scattering (DLS) measurements were executed to elucidate the size of intriguing Mo7–Fe assemblies. The significant scattered intensity increments at the small time-scales shown in Fig. 2e indicated the formation of large structures in the Mo7–$Fe^{3+}$ solution. At the beginning (1 min) of mixing **Mo7** solution and $Fe^{3+}$ cations, the average hydrodynamic radius ($R_h$) of Mo7–Fe assemblies determined by the DLS was -78 nm (Fig. 2f). With time evolution, a new peak representing a larger $R_h$ value appeared while the original one decayed, indicating the formation of larger supramolecular structures in **Mo7** solution with the presence of $Fe^{3+}$.

Considering that this supramolecular interaction occurred between charged species, zeta potential of the Mo7–$Fe^{3+}$ aggregates was conducted. Unexpectedly, we found that the negatively charged $\{$**Mo7**$\}^{6-}$ clusters changed sharply to be positively charged with increasing $Fe^{3+}$ content (Fig. 2g), implying that the outer surface of **Mo7** clusters was rapidly covered by $Fe^{3+}$ cations, leading to a charge neutralization and even a charge reversal, suggesting the association of excess amount of $Fe^{3+}$. Gelation region occurs around the region of neutral charge, suggesting the critical role of electrostatic interaction on the phase transitions. Meanwhile, in the upper solution region, the **Mo7** clusters are clearly positively charged in nature; indicating that the enhanced solubility of **Mo7** due to abundant positive charges is responsible for the upper solution phase.

Isothermal titration calorimetry (ITC) was employed to quantify the interactions between **Mo7** clusters and various cations ($Fe^{3+}$, $Y^{3+}$, $Co^{2+}$ and $Ni^{2+}$), and to clarify why some cations can trigger the gelation while some others cannot. A pronounced enthalpy change was observed in Mo7–Fe system, indicating strong interaction between **Mo7** and $Fe^{3+}$, as shown in Supplementary Fig. 2a and 2e. Slightly weaker interaction between **Mo7** and $Y^{3+}$ was observed than that of **Mo7**/$Fe^{3+}$ (Supplementary Fig. 2a–d), likely because $Y^{3+}$ possesses a larger hydrated ionic size (5.32 Å) than $Fe^{3+}$ (4.80 Å). **Mo7** interacts with $Co^{2+}$ or $Ni^{2+}$ insufficiently, which was suggested by the negligible heat

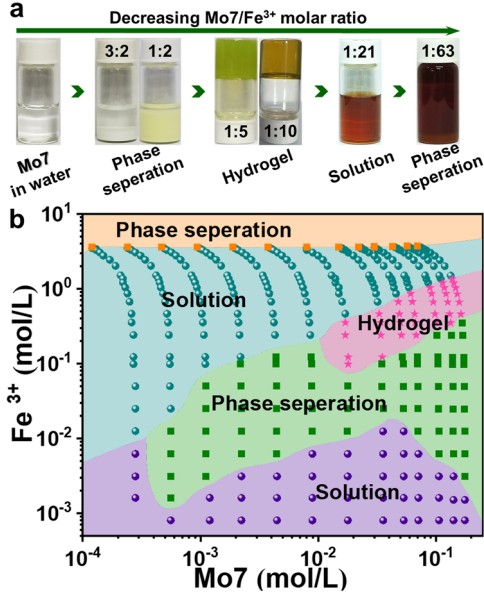

**Fig. 1 | Phase transition. a** Photographs of Mo7–$Fe^{3+}$ systems in aqueous solution at different Mo7/$Fe^{3+}$ molar ratios. **b** Phase diagram of Mo7–Fe system in aqueous solution.

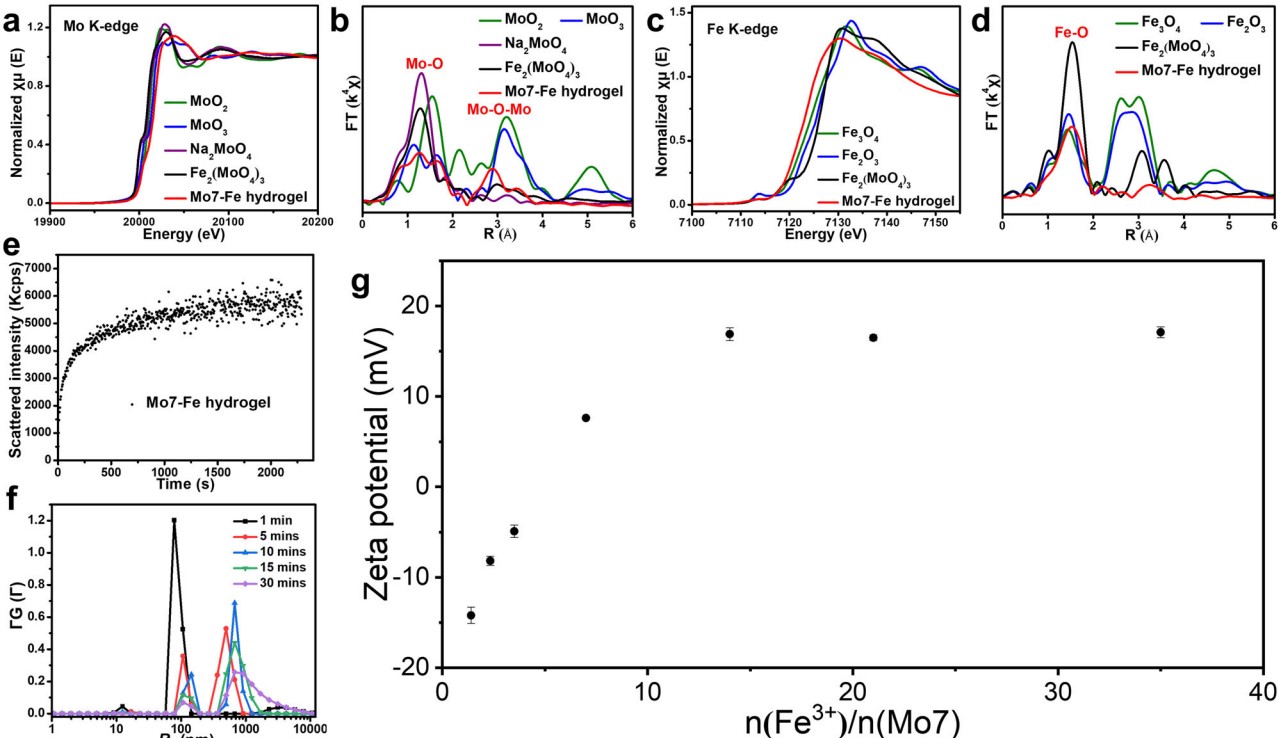

**Fig. 2 | Electrostatic interaction induces Mo7−Fe assemblies. a** Mo K-edge XANES spectra and **b** Fourier transforms of the Mo K-edge EXAFS spectra of Mo7−Fe hydrogel with the references. **c** Fe K-edge XANES spectra and **d** Fourier transforms of the Fe K-edge EXAFS spectra of Mo7−Fe hydrogel with the references. These experimental XAS spectra are shown to prove the electrostatic interaction between **Mo7** clusters and $Fe^{3+}$ ions. **e** Time-resolved scattering intensity plots, representing the increment of the scattered intensity throughout the self-assembly process of Mo7−Fe hydrogel. **f** Time-dependent size distribution by CONTIN analysis of the DLS data. **g** Zeta potentials upon reaction of $6.07 \times 10^{-5}$ mol/L **Mo7** clusters by the adding of $Fe^{3+}$ solution, demonstrating the charge inversion process of **Mo7** anion clusters.

release in Mo7−Co and Mo7−Ni titrations. This explains that **Mo7** is unable to form hydrogels with these two cations even $Co^{2+}$ (hydrated ionic radius of 4.23 Å) and $Ni^{2+}$ (hydrated ionic radius of 4.04 Å) are smaller than that of $Y^{3+}$ (Supplementary Fig. 2e), indicating the importance of cationic valence on the Mo7−cation interaction. However, by adding a small amount of ethanol into Mo7−Co or Mo7−Ni aqueous solutions (Supplementary Fig. 2e), gelation could occur because of the enhanced association of divalent cations around **Mo7**. According to Coulomb's law, when ethanol was added into the aqueous solution containing **Mo7** and $Co^{2+}/Ni^{2+}$, the dielectric constant of the solution is lowered and meanwhile, the electrostatic force is enhanced. When the extent of ion association reaches a certain threshold[11], the Mo7−Fe or Mo7−Y gel network could form spontaneously. It is noted that the typical morphologic characteristics of this percolating ionic gel network are the formation of large branched aggregates, which will be discussed later.

The cryogenic scanning electron microscopy (cryo-SEM) (Fig. 3a) and cryo-TEM images (Fig. 3b) revealed extensive entanglements of uniform long-chain structures of Mo7−Fe aggregates in the hydrogels, which should be responsible for the gelation. The branched structures could be conserved even in the dried Mo7−Fe gel shown by the SEM (Supplementary Fig. 3). The assemblies of Mo7−Fe are further subjected for small-angle X-ray scattering (SAXS) studies for its non-destructive feature during sample characterization. The power law observed at low-q region of SAXS, $I(q) \propto q^{-1.4}$, suggests the feature of semi-flexible long-chain for the Mo7−Fe assemblies (Fig. 3c and Supplementary Fig. 4), similar to the homogalacturonan oligomers with polymerization degrees of 6 and 10[12]. In the high q regime of SAXS curves, the form factor of **Mo7** cluster remains intact in the data of Mo7−Fe hydrogel without crystalline diffraction peaks, implying that **Mo7** clusters are uniformly dispersed in the chain structures.

Meanwhile, in situ optical microscopy studies also conformed these results (Fig. 3d).

To better elucidate the temporal dynamics in the formation of Mo7−Fe aggregates, we performed diffusing wave spectroscopy (DWS) experiment. The dynamic structure factor could not reach zero for gel, indicating the existence of frozen-in concentration fluctuations and non-ergodicity of the systems. The shapes of intensity correlation function (ICF) changed drastically from a stretched exponential to power-law behavior at the gelation threshold (Supplementary Fig. 5)[13]. This verifies the formation of fractal aggregates and loosely tied networks in the Mo7−Fe solution, which is consistent with the model of percolating ionic gel network[11].

Overall, the mechanism of gelation of Mo7−$Fe^{3+}$ in aqueous solution (Fig. 4) can be proposed as follows: (i) the anionic **Mo7** clusters could instantly interact with $Fe^{3+}$ cations to form Mo7−Fe aggregates due to the strong electrostatic attraction; (ii) with the continuous addition of $Fe^{3+}$ cations and $NO_3^-$ anions, the branched Mo7−Fe aggregates overcharged, turned to positively charged, and then rearranged into 1-D fiber-like supramolecular structure, and then formed homogenous Mo7−Fe hydrogels; and (iii) when adding excessive $Fe^{3+}$ and $NO_3^-$, the Mo7−Fe cationic complexes becomes more soluble, making the Mo7−Fe hydrogels redissolved and be back to solution phase.

Molecular Dynamics (MD) simulations have been intensively applied to study the structural properties of the aqueous solutions of $[Mo_7O_{24}]^{6-}$ polyoxometalate[14−17], which provide important insights into the understanding of such relative systems on the molecular level. Herein, coarse-grained MD simulations were performed using LAMMPS[18] to be compared with experimental results; please see the model [Supplementary Fig. 6a] and simulation details in Supporting Information. We coarse-grain the ionic species and water molecules as spherical beads and draw upon the Stockmayer fluid model to

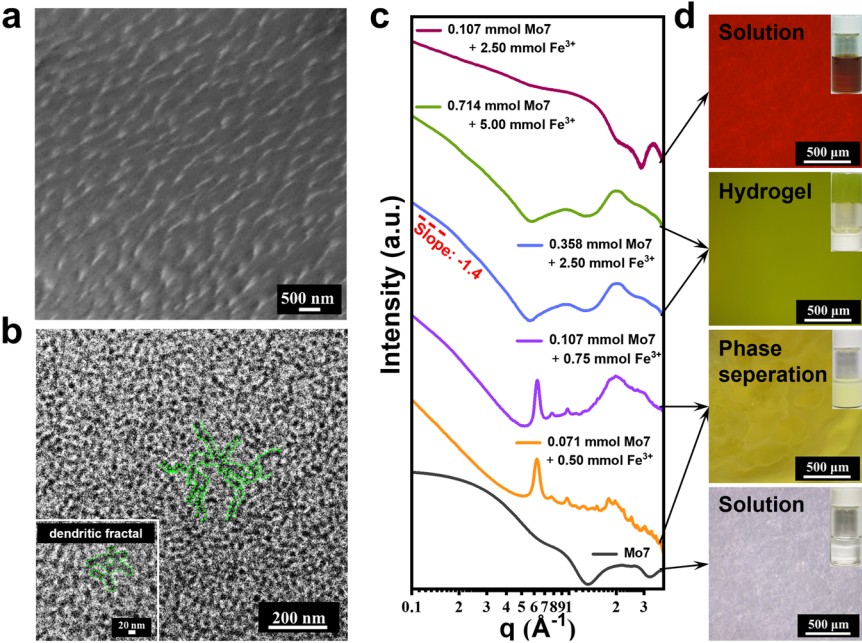

**Fig. 3 | The morphology change of assembles in Mo7−Fe system. a** Cryo-SEM images and **b** cryo-TEM images of Mo7−Fe hydrogel with a **Mo7**/$Fe^{3+}$ molar ratio of 1:10, indicating the formation of branched structures in the Mo7−Fe gel networks. **c** SAXS spectra monitoring the gelation process with different molar ratios of **Mo7**/ $Fe^{3+}$. The arrows are used to connect the SAXS spectra with corresponding phases. **d** In situ optical microscopy of the aggregates formed by **Mo7** and $Fe^{3+}$ in aqueous solution.

consider the dipolar nature of the solvent molecules. This classical model can account for the multi-body correlations among the ion-ion, ion-(dipolar) solvent, and solvent-solvent interactions on relatively large time and length scales, which has proven to be a robust strategy for exploring physicochemical natures of ion-containing mixed solutions[19–21]. We started our analysis from the evolution of the ionic sol–gel transitions. Figure 5a shows the typical simulation snapshots for the ionic sol–gel transition in the $(NH_4)_6[Mo_7O_{24}]$/$Fe(NO_3)_3$ solution at MD time steps of $t = 10^3$, $10^4$, and $10^6$, where the concentrations of $(NH_4)_6[Mo_7O_{24}]$ and $Fe(NO_3)_3$ in the mixed solution are set as $c_1 = 0.2$ [M] and $c_2 = 1.6$ [M], respectively, and the structure factors of ionic species $S_i(q)$ and radial distribution functions of **Mo7** and $NO_3^-$ around each $Fe^{3+}$ during the ionic gel formation are shown in Fig. 5b–d. With the simulation evolution, the peak position of $S_i(q)$ exhibited a significant increase at small $q$, which illustrated the clustering of the ionic species and the formation of the ionic microphase. Accordingly, the significant variations in peaks of $g(r)$ at $r > 4$ Å shown in Fig. 5c, d and Supplementary Fig. 6b illustrated the dominant contribution of the multivalent ions (*i.e.*, **Mo7** and $Fe^{3+}$) as well as the significant amount of monovalent $NO_3^-$ and $NH_4^+$ in the formation of the ionic gels. Further, in Fig. 5e, the calculated diffusion coefficients (D) for various components in the mixed solution revealed that all ionic species apparently slowed down with simulation evolution due to gelation. We then explored the effects of ($c_2$) on the ionic sol–gel transition at a fixed. As shown in Fig. 5f, with increasing $Fe(NO_3)_3$ concentration $c_2$ from $c_2/c_1 = 1$ to 4 at $c_1 = 0.2$ [M], the associated $Fe^{3+}$ ions around each **Mo7** exhibited a gradual increase, until the percolation probability of the ionic species in the mixture increased to 1.0, demonstrating the formation of gel-like structure caused by the enhanced electrostatic correlation between ionic species. The structure factors of ionic species at different ionic compositions (Fig. 5g) show that the multivalent ions (**Mo7** and $Fe^{3+}$) as well as the monovalent $NO_3^-$ ions served as the key components for gel formation. However, in the absence of the plentiful number of $NO_3^-$ ions, the multivalent ions (**Mo7** and $Fe^{3+}$) only tended to form clustering structures as shown by the surge of $S(q)$ at $q \sim 0$, rather than the gelation network structures. This analysis

confirms the role of $NO_3^-$ in bridging the overcharged cationic Mo7−Fe complexes via counterion-mediated attraction, fully consistent with our experimental results.

## Discussion

In summary, we report complex phase transitions involving solution − precipitation − gelation − solution − precipitation, in a POM-based aqueous solution with the addition of simple electrolytes ($Fe(NO_3)_3$). The **Mo7** anions could spontaneously reverse to be positively charged with excessive $Fe^{3+}$. The molecular origin of the gelation is attributed to 1-D branched structures, due to the counterion-mediated attraction between positively charged Mo7−Fe complex and anions ($NO_3^-$). These branched Mo7−Fe micro-aggregates could be rearranged to form gelation or achieve redissolution. To our knowledge, this is the first observation of such complex phase transitions for inorganic ions− demonstrating two distinguished solution phases completely different in nature, separated by a gel phase, all regulated by non-covalent intermolecular interactions. These simple, fundamentally important rules enable us to speculate that similar physical behaviors could be observed in many other pure inorganic solution systems, which could open up new areas in basic research and potential applications.

## Methods

### Reagents

Iron nitrate nonahydrate ($Fe(NO_3)_3 \cdot 9H_2O$), ammonium molybdate tetrahydrate ($(NH_4)_6Mo_7O_{24} \cdot 4H_2O$), and cobalt nitrate hexahydrate ($Co(NO_3)_2 \cdot 6H_2O$) were purchased from Aladdin. Nickel nitrate hexahydrate ($Ni(NO_3)_2 \cdot 6H_2O$) was purchased from Tianjin Yongsheng Fine Chemical Co., Ltd. Yttrium (III) nitrate hexahydrate ($Y(NO_3)_3 \cdot 6H_2O$) was purchased from Macklin. All of them were of analytical reagent grade and used as received without any further purification and deionized water (MillQ, 18.2 MΩ) was used throughout this study.

### Preparation of Mo7−Fe assemblies

In order to obtain Mo7−Fe system, a certain amount of Mo7 was mixed with a certain amount of $Fe^{3+}$ in an aqueous solution. Typically,

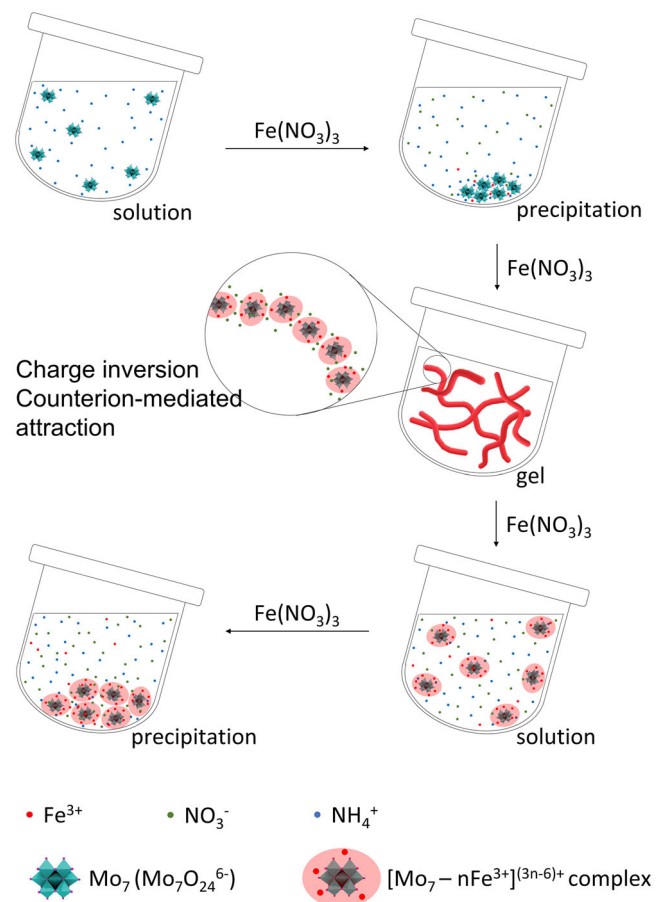

• Fe$^{3+}$     • NO$_3^-$     • NH$_4^+$

Mo$_7$ (Mo$_7$O$_{24}^{6-}$)     [Mo$_7$ − nFe$^{3+}$]$^{(3n-6)+}$ complex

**Fig. 4 | Proposed mechanism.** Schematics of proposed mechanism for gelation and redissolution process in Mo7–Fe system.

Fe(NO$_3$)$_3$·9H$_2$O (1444 mg) and (NH$_4$)$_6$Mo$_7$O$_{24}$·4H$_2$O (442 mg) were dissolved in 2.5 mL of deionized water, respectively. The mixture of the above solution was stirred for 10 min, leading to the formation of a yellow clear solution at room temperature. The solution was left for 3 mins to form the Mo7–Fe hydrogel within the system.

## Dynamic shear rheology
Rheological measurements were performed on a rotated rheometer (AR 2000ex, TA Instrument, America). Time-sweep tests were performed at 25 °C with the frequency and strain fixed at 10 Hz and 2%, respectively.

## X-ray absorption spectroscopy (XAS)
The Mo and Fe K-edge X-ray absorption fine structure was collected at the 1W1B beamline of the Beijing Synchrotron Radiation Facility.

## Static light scattering (SLS) and dynamic light scattering (DLS)
A commercial Brookhaven Instrument light scattering spectrometer with a solid-state laser (OBIS LX 633 nm 70 mW) and a BI 9000AT digital correlator, was used for both the static light scattering (SLS) and dynamic light scattering (DLS) measurements at a fixed angle (90). The temperature of the sample chamber was maintained at room temperature during the measurements. The intensity-intensity time correlation functions were analysed by the constrained regularized (CONTIN) method. The average apparent translational diffusion coefficient, $D$, was determined from the normalized distribution function of the characteristic line width, Γ(G). The average hydrodynamic radius ($R_h$) was converted from $D$ through the Stokes-Einstein equation: $R_h = kT/6\pi\eta D$, where $k$ was

the Boltzmann constant and $\eta$ was the viscosity of the solvent at temperature ($T$).

Non-ergodic gels were analysed by DLS, and the asymptotic behavior of time-intensity correlation function near the gelation point could be described by the following functions:

$$g_T^{(2)}(\tau) - 1 \approx \sigma_I^2 \{A \exp(-Dq^2\tau) + (1-A) \exp[-(\tau/\tau_C)^\beta]\}^2 \text{ (sol)}$$

$$g_T^{(2)}(\tau) - 1 \approx \sigma_I^2 \{A \exp(-Dq^2\tau) + (1-A)[1 + (\tau/\tau^*)]^{(n-1)/2}\}^2 \text{ (gel point)}$$

where $\sigma_I^2$, the initial amplitude of ICF; $A$, the fraction of the collective diffusion mode; $D$, the diffusion coefficient of the fast mode; $\tau_c$, the characteristic time for the stretched exponential mode; $\beta$, stretched exponent ($0 < \beta < 1$); $\tau^*$, the characteristic time where the power law behavior appears; and $n$, the fractal dimension of scattered photons ($0 < n < 1$).

## Zeta potential analysis
A Brookhaven Instruments commercial *ZetaPALS* analyzer was used to measure the zeta potential of Mo7–Fe assemblies in sample solutions. The analyzer was equipped with a 35-mW solid-state laser operating at 660 nm. According to the instrument design, particles with diameters from 10 nm to 30 μm (depending on particle density) and zeta potential ranging from −150 to +150 mV could be measured, and the data accuracy and repeatability were both ±2% for dust-free samples.

## Isothermal titration calorimetry (ITC)
ITC measurements were conducted using a standard volume Nano ITC isothermal titration calorimeter (TA Instruments) with a 1.0 mL Hastelloy C sample cell and 250 μL syringe. The experiments were carried out at 25 °C by using 25 consecutive injections of 10 μL. A rotation rate of 250 rpm was used to ensure the proper, continuous mixing of solutions in the chamber. In all experiments, the background heat was subtracted to reach the final results. Fitting was performed using NanoAnalyze software provided by TA Instruments. The independent binding site model was applied to analyze the results which provided the binding number and interaction energy between Mo7 and different salts.

## Electron microscope (EM) analysis
Optical images were performed on a Zeiss Microscopy with a system control panel (SYCOP 3). The cryogenic scanning electron microscopy (cryo-SEM) images of Mo7–Fe hydrogel were observed using Hitachi Regulus 8100 scanning electron microscopy (SEM). Scanning electron microscope (SEM) images of dried Mo7–Fe gel were obtained from the field emission SEM (Hitachi S-4800, Japan). The hydrogel was flash-frozen in liquid nitrogen and dried in a freeze dryer overnight. The cryogenic transmission electron microscopy (cryo-TEM) images of Mo7–Fe hydrogel were observed using JEOL JEM-3200FSC.

## Small-angle X-ray scattering (SAXS)
SAXS data were collected using a two-dimensional wide-angle X-ray diffraction instrument equipped with the Home Lab system of Rigaku. For monodisperse interacting particles, the total scattering intensity was a function of their single-particle form factor $P(q)$ and structure factor $S(q)$ and could be written as:

$$I(q) = n_m \Delta\rho^2 V^2 p(q) S(q) + I_B$$

where the scattering intensity $I(q)$ was measured as a function of scattering vector $q$; $n_m$, the number density of scattering particles; $\Delta\rho$, the difference of scattering length density of particles from the solvent; $V$, the scattering particle volume; $P(q)$, the form factor which

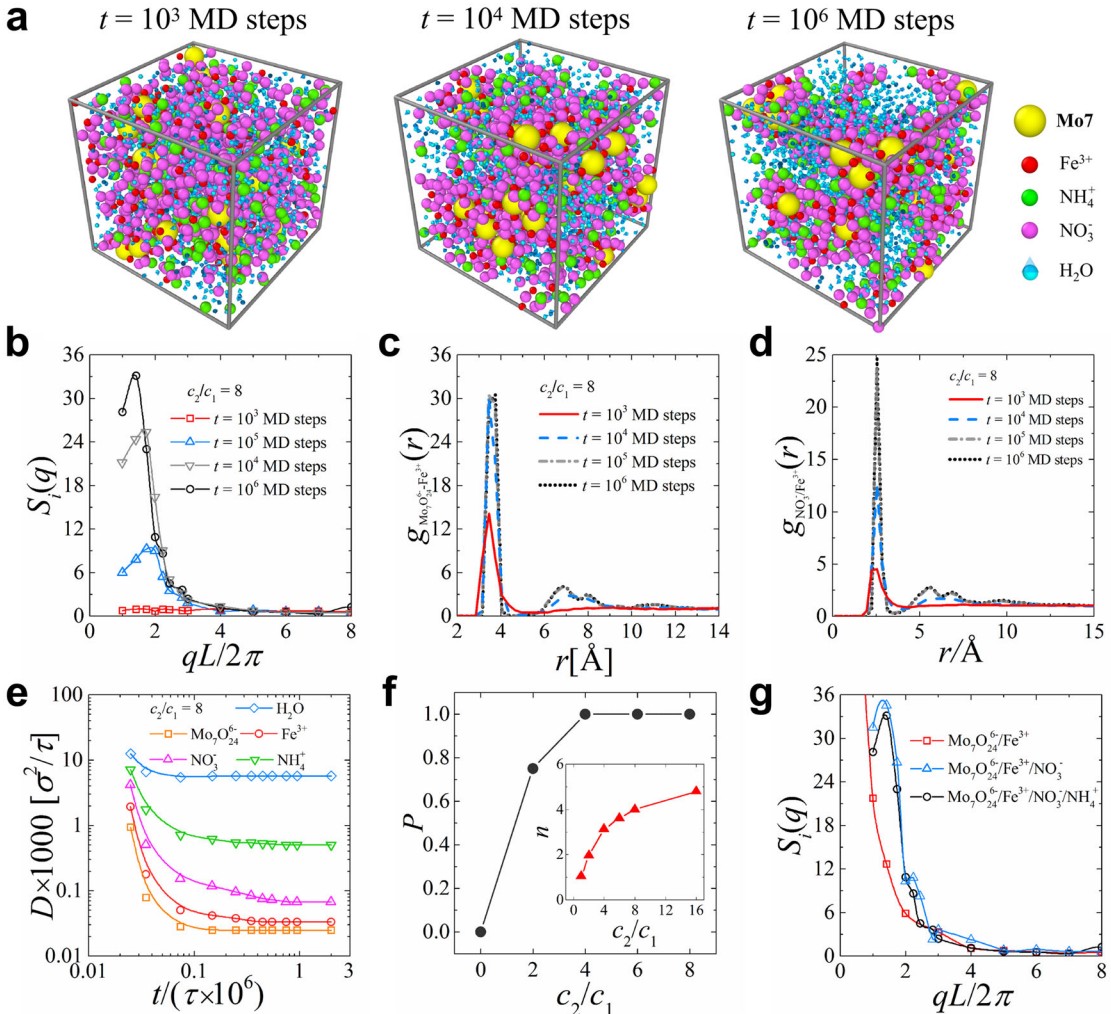

**Fig. 5 | The molecular dynamics (MD) simulations of Mo7–Fe system. a** Typical simulation snapshots for the ionic sol–gel transition in the $(NH_4)_6[Mo_7O_{24}]$/$Fe(NO_3)_3$ solution at different MD time steps. The concentrations of $(NH_4)_6[Mo_7O_{24}]$ and $Fe(NO_3)_3$ are set as $c_1 = 0.2$ [M] and $c_2 = 1.6$ [M], respectively. **b** Structure factors of ionic species $S_i(q)$, **c** radial distribution functions of **Mo7** around $Fe^{3+}$ and **d** radial distribution functions of $NO_3^-$ around $Fe^{3+}$ at different simulation time steps. **e** Diffusion coefficients of each component ($D$) during the ionic gel formation. **f** Percolation probability of ionic species ($P_i$) and number of neighboring $Fe^{3+}$ ions around each **Mo7** as a function of concentration ratio $c_2/c_1$. **g** Structure factors of ionic species at different ionic compositions.

depends on the shape and size of scattering particles; $S(q)$, the structure factor, accounting for the inter-particle interactions between the neighboring scattering particles; and $I_B$, the background intensity.

The net scattering intensity of the particles $I_r(q)$ could be derived from the total observed scattering intensity ($I(q)$) by subtracting background scattering for the following equation:

$$I_r(q) = I(q) - I_B$$

where $I_B$ was the scattering intensity of the solvent-filled container.

**Diffusing wave spectroscopy (DWS)**
The DWS experiments were performed in backscattering geometry. Diode laser with high coherence length and low noise ($\lambda = 685$ nm, output power 40 mW, LGK 7665, LASOS Lasertechnik GmbH) was expanded is expanded by a lens so that it illuminates an area of incident face that is wider than $l^*$. The central part of the beam was uniform illumination on the curvette side. The laser beam impinged perpendicularly on the side of the curvette. The hydrogel was contained in quartz cuvettes of thickness 10 mm.

## Data availability
The experiment data that support the findings of this study are available within the article and its Supplementary Information file. Source data are provided with this paper.

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

## Acknowledgements

The authors gratefully acknowledge the financial support from the National Natural Science Foundation of China (Nos. 21975097, 21871042, 21673098, 51975245 and 22073094), Key Science and Technology R&D Projects of Jilin Province (2020C023-3), Program of Jilin University Science and Technology Innovative Research Team (2020TD-03), Natural Science Foundation of Jilin Province (Grant No. 20200201083JC), Jilin Provincial Education Department (Grant No. JJKH20201169KJ), the Jilin Provincial Science & Technology Department (20190303039SF, 20210402059GH), the Science and Technology Plan Projects of Yunnan Province (No. 202101BC070001-007) and the Major Science and Technology Projects for Independent Innovation of China FAW Group Co., Ltd (No. 20220301018GX). T.L. acknowledges support from the National Science Foundation (NSF CHE1904397 and EFRI E3P2132178) and The University of Akron.

## Author contributions

Z.W., S.L., T.L., and H.Z. conceived and designed the study. S.L., Z.L., T. L. and H.Z. supervised the project. Z.W., Y.Y., and P.Y. conducted the experiments and analyzed the data. X.D. performed the MD simulations. Z.W., S.L., X.D., T.L. and H.Z. wrote the manuscript. All of the authors provided input and reviewed the manuscript.

## Competing interests

The authors declare no competing interests.
