## [Peer Review File · Nature Communications]

REVIEWER COMMENTS

Reviewer #1 (Remarks to the Author):

The present manuscript reports the investigation of complex phase behaviour involving multiple phase transitions discovered in a $[\text{Mo}_7\text{O}_{24}]^{6-}/\text{Fe}^{3+}$ aqueous solution. The authors confirmed by experimental data and molecular dynamics simulations the formation of linear/branched supramolecular structures driven by counterion electrostatic interactions in the case of poorly charged POM cluster. The sequential concentration dependent and electrostatically driven phase transitions is a very interesting behaviour the small POM species under investigation. The preparation and characterisation of the solution is in general clear, and the manuscript is in well written and referenced. .

Page 3, below Fig1: "With the titration of more $\text{Fe}(\text{NO}_3)_3$ (e.g., ~ 0.2 mol/L for 0.04 mol/L Mo_7), solid precipitates disappeared and the solution turned to a transparent yellow-green hydrogel phase." Two lines before that the concentrations used were ~ 0.4 mol/L for 0.02 mol/L Mo_7 . This is confusing. When the gel forms? Upon increase or decrease of $[\text{Fe}]$? Please correct or provide clarification.

Page 5, top of the page: There was no change in the oxidation state of both Mo and Fe throughout the series of transitions, ..." It is true that any reduction of Mo would have been obvious due to the usually intense colour in partially reduced Mo centres. However, the colour of the solution when the hydrogel is formed (Fig. 1) has an olive green colour which is typical for Fe^{2+} in aqueous solutions which then changes to brown in the solution phase which is typical for Fe^{3+} . The authors provide the Fe K-edge using the $\text{Fe}_2(\text{MoO}_4)_3$ compound for comparison, however, the Fe in this study the transition metal exists as a free hydrated centre in solution and interacts only via electrostatic interactions. The authors need to provide some clarity at this point since the claim that the oxidation state for Fe doesn't change might be confusing for the readers.

Page 5, end of first paragraph: The authors state: "Meanwhile, the hydrogel is also observed when titrating $\text{Y}(\text{NO}_3)_3$ into Mo_7 solution, while Co^{2+} and Ni^{2+} cannot trigger gelation." What exactly happens in the presence of divalent cations? The authors could a bit further this part specifically when later in the manuscript the divalent ions do form hydrogels in a lower polarity solvent.

Since the authors are working in aqueous solutions it would be interesting to discuss what is the pH value of the reaction mixture specifically upon consecutive addition of $\text{Fe}(\text{NO}_3)_3$.

The manuscript presents a very interesting phenomenon of phase transition and gel formation of POM species in the presence of non-covalent interactions with transition metals which to my knowledge hasn't been observed before. My opinion is that provides important clues on underlying processes that take place in the background which are important for the assembly processes that govern the formation of new nanosized clusters and are crucial in polyoxometalate chemistry. I have the feeling that the manuscript can be accepted for publication in Nature Commun after some minor revisions in order to improve clarity and avoid misunderstandings for the readers.

Reviewer #2 (Remarks to the Author):

The manuscript reports the unprecedented discovery of complex phase behaviour of aqueous solutions of a $[\text{Mo}_7\text{O}_{24}]^{6-}$ polyoxometalate with the continuous addition of Fe^{3+} . Multiple phase transitions were observed in the experiments, which were characterised by a bunch of analytical techniques and also by molecular dynamics (MD) simulations. Regarding this part, the use of MD

to study the aggregation of polyoxometalates has been studied by several authors and reported in literature. See for instance:

J. Phys. Chem. B 2016, 120, 12959–12971
J. Phys. Chem. B 2019, 123, 10505–10513
Inorg. Chem. 2019, 58, 3881–3894
Chem. Sci., 2020, 11, 11072

Although those simulations were performed using atomistic models and smaller simulation boxes, Figure 10 from Solé-Daura et al. (Inorg. Chem. 2019) for instance shows a schematic view of a phenomenon closely related to the experiments described in the manuscript. Readers would appreciate a discussion of the advantages of using a coarse-grained models instead of detailed atomistic potentials. The shape of such a kind of macroanions some times differ from ideally spherical beds, so the coarse-grained model might not be describe specific close contacts anion-cation at the anion surface. Could the authors comment also on the validation of the potentials used?

Then, it seems the some MD part is missing in the SupInfo. In page 9, first paragraph: "please see the model and simulation details in Supporting Information." But in the SupInfo there is only Figure S6 for the MD part. Figure S6 caption is incomplete, and the Figure labels look erroneous (RDF Mo7-NO₃, but labeled Mo7-NH₄) ????

Other comments:

- Figure 1 photograph is not well sorted by increasing Fe³⁺ concentration.
- Figure 1. Down part description is missing in the figure caption.
- Some typos and incomplete sentences in the MD part.
- Some references badly placed within the text.

Overall, in my opinion, the experiments reported here are unprecedented so they deserve publication. However, the manuscript is still incomplete and needs revision.

RESPONSE TO REVIEWERS' COMMENTS

Reviewer #1 (Remarks to the Author):

The present manuscript reports the investigation of complex phase behaviour involving multiple phase transitions discovered in a $[\text{Mo}_7\text{O}_{24}]^{6-}/\text{Fe}^{3+}$ aqueous solution. The authors confirmed by experimental data and molecular dynamics simulations the formation of linear/branched supramolecular structures driven by counterion electrostatic interactions in the case of poorly charged POM cluster. The sequential concentration dependent and electrostatically driven phase transitions is a very interesting behaviour the small POM species under investigation. The preparation and characterisation of the solution is in general clear, and the manuscript is in well written and referenced.

Question 1: Page 3, below Fig1: “With the titration of more $\text{Fe}(\text{NO}_3)_3$ (e.g., ~ 0.2 mol/L for 0.04 mol/L Mo7), solid precipitates disappeared and the solution turned to a transparent yellow-green hydrogel phase.” Two lines before that the concentrations used were ~ 0.4 mol/L for 0.02 mol/L Mo7. This is confusing. When the gel forms? Upon increase or decrease of $[\text{Fe}]$? Please correct or provide clarification.

Response:

We thank the Reviewer for the careful reading of our manuscript, and apologize for the confusion regarding concentration range. For “Two lines before that the concentrations used”, the concentration was for precipitates formation. To avoid confusion, we gave a specific concentration example for the precipitates formation with 0.02 mol/L Mo7 and 0.01 - 0.04 mol/L $\text{Fe}(\text{NO}_3)_3$. By increasing the concentration of $\text{Fe}(\text{NO}_3)_3$ to 0.1~0.2 mol/L, the gel forms. In response to this comment, we have revised the manuscript by adding the following text on page 4:

“In a clear, dilute 2.5 mL aqueous solution of $(\text{NH}_4)_6[\text{Mo}_7\text{O}_{24}]$ (0.02 mol/L), precipitation is observed when adding 2.5 mL of $\text{Fe}(\text{NO}_3)_3$ (0.01 mol/L~0.04mol/L), indicating that the solubility of Mo7 decreases with the presence of $\text{Fe}(\text{NO}_3)_3$ (pH~0.9). With the titration of more $\text{Fe}(\text{NO}_3)_3$ (0.1~0.2 mol/L), solid precipitates disappeared and the solution turned to a hydrogel phase. The hydrogel phase can form within the concentration ratio of Mo7: Fe^{3+} ranging from 1:5 to 1:10 (pH~0.3).”

Question 2: Page 5, top of the page: There was no change in the oxidation state of both Mo and Fe throughout the series of transitions, ...” It is true that any reduction of Mo would have been obvious due to the usually intense colour in partially reduced Mo centres. However, the colour of the solution when the hydrogel is formed (Fig. 1) has an olive green colour which is typical for Fe^{2+} in aqueous solutions which then changes to brown in the solution phase which is typical for Fe^{3+} . The authors provide the Fe K-edge using the $\text{Fe}_2(\text{MoO}_4)_3$ compound for comparison, however, the Fe in this study the transition metal exists as a free hydrated centre in solution and interacts only via electrostatic interactions. The authors need to provide some clarity at this point since the claim that the oxidation state for Fe doesn't change might be confusing for the readers.

Response: We completely agree with the Reviewer. The olive green colour gel is due to partial reduction of Fe^{3+} to Fe^{2+} . In the study of the phase transition process with a Mo7: Fe^{3+} molar ratio of 1:5, abundant NH_4^+ are present ($\text{NH}_4^+:\text{Fe}^{3+}$ molar ratio 4:5). Part of Fe^{3+} ions might be reduced to Fe^{2+} by NH_4^+ . Since the gel is not clear enough, we explored more parameters and obtained the transparent, yellow gel at molar ratio Mo7: Fe^{3+} = 1:10. The XPS data for this gel show that iron was in the state of Fe^{3+} . Please refer to the figure below.

Fig. S1a The XPS spectra for Fe 2p, which shows 100% Fe³⁺.

In response to this comment, we have updated Fig. S1 (a) and revised the manuscript as follows: “The hydrogel phase can form within the concentration ratio of Mo7:Fe³⁺ ranging from 1:5 to 1:10 (pH~0.3). However, when the concentration ratio of Mo7:Fe³⁺ was 1:5, the gel in olive green color was not transparent. We assumed that part of Fe³⁺ was reduced to Fe²⁺ due to NH₄⁺. To obtain the homogeneous transparent gel, we further explored more parameters and obtained the transparent yellow gel with Mo7:Fe³⁺ being 1:10. The XPS data for Fe 2p orbital is divided into three integrated peaks, the peaks at 712.2 and 726.1 eV belong to Fe 2p_{3/2} and Fe 2p_{1/2} of Fe³⁺, respectively (Fig.S1a).”

Question 3: Page 5, end of first paragraph: The authors state: “Meanwhile, the hydrogel is also observed when titrating Y(NO₃)₃ into Mo₇ solution, while Co²⁺ and Ni²⁺ cannot trigger gelation.” What exactly happens in the presence of divalent cations? The authors could a bit further this part specifically when later in the manuscript the divalent ions do form hydrogels in a lower polarity solvent.

Response:

We thank the Reviewer for the insightful comment.

In an aqueous solution, Co²⁺ and Ni²⁺ cannot trigger the gel formation. Based on Coulomb's law, the electrostatic force is proportional to the charge amount of two ions. We assumed that due to the less charge amount, the divalent Co²⁺ and Ni²⁺ ions can hardly form gel with Mo7 and the aqueous solution remains in its liquor state.

$$F = \frac{q_1 q_2 e^2}{4\pi\epsilon_0\epsilon_r r^2}$$

However, when ethanol was added into the aqueous solution containing Mo7 and Co²⁺/Ni²⁺, the dielectric constant ϵ_r is lowered, and meanwhile, the electrostatic force is increased, which results in the gel formation. We used this comparative experiment to demonstrate that electrostatic interactions serve as the main drive for gel formation.

In response to this comment, we have revised the manuscript by adding the following texts:

“Meanwhile, the hydrogel is also observed when titrating Y(NO₃)₃ into Mo₇ aqueous solution, while Co²⁺ and Ni²⁺ cannot trigger gelation and the aqueous solution remains in its liquor state. As indicated by Coulomb's law, the electrostatic force is proportional to the charge of two ions. We

assumed that due to the less charge (compared to Fe^{3+}), the divalent Co^{2+} and Ni^{2+} ions cannot form gel with Mo7 in an aqueous solution.”

“However, by adding a small amount of ethanol into Mo7-Co or Mo7-Ni aqueous solutions (Supporting Information Fig. S2e), gelation could occur because of the enhanced association of divalent cations around Mo7. According to Coulomb's law, when ethanol was added into the aqueous solution containing Mo7 and $\text{Co}^{2+}/\text{Ni}^{2+}$, the dielectric constant of the solution is lowered and meanwhile, the electrostatic force is enhanced.”

Question 4: Since the authors are working in aqueous solutions it would be interesting to discuss what is the pH value of the reaction mixture specifically upon consecutive addition of $\text{Fe}(\text{NO}_3)_3$.

Response: We thank the Reviewer for the constructive suggestion. As shown in our answer to Question 1, we have discussed the pH value for the phase transformation process of Mo7-Fe in the manuscript. The overall pH is very low. When precipitate occurred, pH was at about 0.9. When the gel formed, pH was about 0.3. When the gel transformed into solution, pH is approaching 0.

The manuscript presents a very interesting phenomenon of phase transition and gel formation of POM species in the presence of non-covalent interactions with transition metals which to my knowledge hasn't been observed before. My opinion is that provides important clues on underlying processes that take place in the background which are important for the assembly processes that govern the formation of new nanosized clusters and are crucial in polyoxometalate chemistry. I have the feeling that the manuscript can be accepted for publication in Nature Commun after some minor revisions in order to improve clarity and avoid misunderstandings for the readers.

Response: We thank the Reviewer for the overall positive evaluation of our work, and sincerely appreciate the Reviewer for the constructive suggestions and insightful comments.

Reviewer #2 (Remarks to the Author):

The manuscript reports the unprecedented discovery of complex phase behaviour of aqueous solutions of a $[\text{Mo}_7\text{O}_{24}]^{6-}$ polyoxometalate with the continuous addition of Fe^{3+} . Multiple phase transitions were observed in the experiments, which were characterised by a bunch of analytical techniques and also by molecular dynamics (MD) simulations.

Question 1: Regarding this part, the use of MD to study the aggregation of polyoxometalates has been studied by several authors and reported in literature. See for instance:

J. Phys. Chem. B 2016, 120, 12959–12971

J. Phys. Chem. B 2019, 123, 10505–10513

Inorg. Chem. 2019, 58, 3881–3894

Chem. Sci., 2020, 11, 11072

Although those simulations were performed using atomistic models and smaller simulation boxes, Figure 10 from Solé-Daura et al. (Inorg. Chem. 2019) for instance shows an schematic view of a phenomenon closely related to the experiments described in the manuscript. Readers would appreciate a discussion of the advantages of using a coarse-grained models instead of detailed atomistic potentials. The shape of such a kind of macroanions some times differ from ideally spherical beds, so the coarse-grained model might not be describe specific close contacts anion-cation at the anion surface. Could the authors comment also on the validation of the potentials used?

Response:

We thank the Reviewer for the insightful comment.

We completely agree with the Reviewer that atomistic simulations serve as an effective approach to exploring the physicochemical natures of aqueous solutions of $[\text{Mo}_7\text{O}_{24}]^{6-}$ polyoxometalate, and several pioneering simulation studies (such as those references the referee mentioned) have provided important insights into the deep understanding of such relative systems. We have included these references into the manuscript.

While atomistic simulations are very useful to study the polyoxometalate solutions, the atomistic simulation methods generically still have some limited accessibility for systems of long time scale and large length scale (such as the process of sol-gel transitions and phase transitions), and the accuracy of the results could be dependent on the choice of specific molecular force fields. The vastly intensive computation is usually a practical issue in explaining numerous diverse experimental data that come one after another, and the fact that simulation often requires a large set of model parameters routinely imposes researchers on a daunting task. On the contrary, as well-accepted, the coarse-grained simulations in statistical mechanics typically consist of a minimal set of model parameters and they are, therefore, also considered a robust strategy that is well suited for exploring the soft-condensed matter systems [Cisneros, Karttunen, Ren, Sagui, Chem. Rev. 114, 779-814, (2014)].

In this work, we aim to systematically explore the structural properties of $\text{NH}_4\text{Mo}_7\text{O}_{24}/\text{Fe}(\text{NO}_3)_3$ mixed solutions at different Mo_7 concentrations, and particularly, the sol-gel transitions that may occur in a relatively larger time and length scale. In our coarse-grained model, the ions and solvent molecules are coarse-grained as spherical particles and the Stockmayer fluid model is drawn to consider the dipolar nature of the solvent molecules. Given that the electrostatic correlations and ionic solvation serve as the main driving force for the complex phase transition discussed in the current work, our coarse-grained model can effectively capture the coupling of these interactions. By neglecting the detailed molecular structures, our simulations are independent of any atomistic force field and can reflect the structural evolutions in relatively large time and length scales. Therefore, we think our systematic coarse-grained simulations could be suitable for studying the sol-gel transitions of $\text{NH}_4\text{Mo}_7\text{O}_{24}/\text{Fe}(\text{NO}_3)_3$ mixed solutions and could be used to clarify the underlying mechanism on the molecular level.

In response to the Reviewer's comment, we have revised the manuscript and Supporting Information by adding the new texts:

“Molecular Dynamics (MD) simulations have been intensively applied to study the structural properties of the aqueous solutions of $[\text{Mo}_7\text{O}_{24}]^{6-}$ polyoxometalate²⁷⁻³⁰, which provide important insights into the understanding of such relative systems on the molecular level. Herein, coarse-grained MD simulations were performed using LAMMPS³¹ to be compared with experimental results; please see the model [Fig. S6 (a)] and simulation details in Supporting Information. We coarse-grain the ionic species and water molecules as spherical beads and draw upon the Stockmayer fluid model to consider the dipolar nature of the solvent molecules. This classical model can account for the multi-body correlations among the ion-ion, ion-(dipolar) solvent, and solvent-solvent interactions on relatively large time and length scales, which has proven to be a robust strategy for exploring physicochemical natures of ion-containing mixed solutions³²⁻³⁴.”

“The coarse-grained simulations in statistical mechanics typically consist of a minimal set of model parameters and they are, therefore, considered a robust strategy that is well suited for exploring the ion-containing systems.”

“Given that the electrostatic correlations and ionic solvation serve as the main driving force for the complex phase transition discussed in the current work, our coarse-grained model can effectively capture the coupling of these interactions. By neglecting the detailed molecular structures, our simulations are independent of any atomistic force field and can reflect the structural evolutions in relatively large time and length scales. Therefore, the systematic coarse-grained simulations are suitable for studying the sol-gel transitions of $\text{NH}_4\text{Mo}_7\text{O}_{24}/\text{Fe}(\text{NO}_3)_3$ mixed solutions and could be used to clarify the underlying mechanism on the molecular level.”

Question 2: Then, it seems the some MD part is missing in the SupInfo. In page 9, first paragraph: "please see the model and simulation details in Supporting Information." But in the SupInfo there is only Figure S6 for the MD part. Figure S6 caption is incomplete, and the Figure labels look erroneous (RDF Mo7-NO3, but labeled Mo7-NH4) ????

Response:

We thank the Reviewer for the careful reading of our manuscript.

In our previous manuscript, we have indeed missed the model and simulation details in the Supporting Information and have actually included this part in the Methods Section. In response to the Reviewer's comment, we have revised the manuscript by adding a schematic illustration of the coarse-grained simulation model and simulation details in the Supporting Information.

Additionally, as pointed out by the Reviewer, in our previous manuscript, the label and caption in Fig. S6 (b) are inconsistent, which shows the radial distribution functions of Mo7 around NH_4^+ at different simulation time steps. We have corrected this typo.

Other comments:

- Figure 1 photograph is not well sorted by increasing Fe^{3+} concentration.

- Figure 1. Down part description is missing in the figure caption.

Response: We thank the Reviewer for the careful reading of our manuscript. We have revised the figure description by replacing “Increasing $\text{Mo7}/\text{Fe}^{3+}$ molar ratio” with “Decreasing $\text{Mo7}/\text{Fe}^{3+}$ molar ratio” and added the caption for the down part of Figure 1.

- Some typos and incomplete sentences in the MD part.

- Some references badly placed within the text.

Response: We thank the Reviewer for the careful reading of our manuscript. We have carefully checked and revised the MD part and rearranged the inappropriately placed references.

Overall, in my opinion, the experiments reported here are unprecedented so they deserve publication. However, the manuscript is still incomplete and needs revision.

Response: We thank the Reviewer for the overall positive evaluation of our work, and sincerely appreciate the Reviewer for the constructive suggestions and insightful comments.

REVIEWERS' COMMENTS

Reviewer #1 (Remarks to the Author):

The authors have addressed satisfactorily the reviewers' concerns and improved the overall quality and clarity of the manuscript. I am happy to recommend the publication of the manuscript in its current state.

Reviewer #2 (Remarks to the Author):

This reviewer appreciates author's responses and clarifications.

Publish as it is.